# A Molecular Perspective and Role of NAD^+^ in Ovarian Aging

**DOI:** 10.3390/ijms25094680

**Published:** 2024-04-25

**Authors:** Mehboob Ahmed, Umair Riaz, Haimiao Lv, Liguo Yang

**Affiliations:** 1Hubei Hongshan Laboratory, Wuhan 430070, China; mehboob.alyani@gmail.com (M.A.); umair.riaz@iub.edu.pk (U.R.); lvhaimiao0310@webmail.hzau.edu.cn (H.L.); 2Key Laboratory of Animal Genetics, Breeding and Reproduction, Ministry of Education, College of Animal Science and Technology, Huazhong Agricultural University, Wuhan 430070, China; 3National Center for International Research on Animal Genetics, Breeding and Reproduction (NCIRAGBR), Ministry of Science and Technology, Huazhong Agricultural University, Wuhan 430070, China

**Keywords:** ovarian aging, NAD^+^ metabolism, aging hallmarks, female fertility

## Abstract

The decline in female fecundity is linked to advancing chronological age. The ovarian reserve diminishes in quantity and quality as women age, impacting reproductive efficiency and the aging process in the rest of the body. NAD^+^ is an essential coenzyme in cellular energy production, metabolism, cell signaling, and survival. It is involved in aging and is linked to various age-related conditions. Hallmarks associated with aging, diseases, and metabolic dysfunctions can significantly affect fertility by disturbing the delicate relationship between energy metabolism and female reproduction. Enzymes such as sirtuins, PARPs, and CD38 play essential roles in NAD^+^ biology, which actively consume NAD^+^ in their enzymatic activities. In recent years, NAD^+^ has gained much attention for its role in aging and age-related diseases like cancer, Alzheimer’s, cardiovascular diseases, and neurodegenerative disorders, highlighting its involvement in various pathophysiological processes. However, its impact on female reproduction is not well understood. This review aims to bridge this knowledge gap by comprehensively exploring the complex interplay between NAD^+^ biology and female reproductive aging and providing valuable information that could help develop plans to improve women’s reproductive health and prevent fertility issues.

## 1. Introduction

The correlation between advancing chronological age and declining female fecundity (the ability to produce offspring) has been well established and widely recognized [1]. Over the past 150 years, a notable increasing trend in life expectancy has been observed in developed nations [2]. However, in contrast, the timing of reproductive senescence, specifically the age at natural menopause (ANM), has remained remarkably stable without significant changes [3]. The increasing involvement of women in education and the workforce has led to a substantial postponement of childbearing. The availability of reliable and FDA-approved contraception methods and societal change have empowered women to take control of their reproductive choices and pursue higher education and careers [4]. Nevertheless, this trend of delayed childbearing poses a unique challenge for females. In contrast to the male reproductive system, which can continually produce spermatogonial cells, females have a predefined number of oocytes [5,6]. These oocytes constitute a finite pool of primordial follicles in the ovary, often called the ovarian reserve (OR). This reserve gradually diminishes in quantity and quality throughout the lifetime, significantly impacting reproductive efficiency and aging [7]. Although female fertility peaks around the age of 25, a decline in fecundity starts to occur after the age of 30 [8]. Surprisingly, women aged 35–39 have 31% lower fecundity than those aged 20–24 [9]. The genetic integrity of oocytes declines with age, leading to a cessation of natural fertility around a decade before menopause. Cellular hallmarks of aging manifest in the ovary before reproductive aging, and the decline in OR with advancing age is associated with reduced fertility [10].

The aging process is typically connected with declines in various biological systems, with one of the initial systems affected being the female reproductive system. Nicotinamide adenine dinucleotide (NAD^+^), an essential coenzyme involved in many cellular processes, especially those critical for female reproduction, could be one reason for this drop. NAD^+^ is fundamentally connected to energy production, which is essential to support the growth and development of oocytes. Besides this, NAD^+^ is critical for DNA repair and cell signaling, making it indispensable for oocyte development, embryo implantation, and the overall maintenance of the uterine environment. Sirtuins, PARPs, and CD38 are the essential enzymes that play important roles in the complex web of NAD^+^ biology. These enzymes consume NAD^+^ in their enzymatic activities, and their roles become particularly significant in aging and age-related diseases. The activity of sirtuins is compromised with advanced age, resulting in the dysregulation of cellular processes and potentially impacting ovarian aging. PARPs and CD38 have been shown to exhibit increased activity with age, leading to accelerated NAD^+^ consumption.

Increasing NAD^+^ levels have offered advantages in mitigating the effects of aging and age-related diseases. Many studies have evaluated its impact on neurodegenerative diseases, cardiovascular diseases, metabolic diseases, and cancer [11,12,13,14]. In recent years, NAD^+^ biology has garnered significant attention, prompting the publication of numerous informative review articles exploring its role in various aspects of age-related diseases and longevity [15,16,17,18]. Despite these advancements, its influence on ovarian aging remains to be elucidated. This review article investigates the intricate molecular mechanisms governing ovarian aging. It unveils the biological significance of NAD^+^ in female reproduction, shedding light on how NAD^+^ homeostasis impacts the reproductive aging process. In addition, we look into how sirtuins, PARPs, and CD38 use NAD^+^ and how that might affect female reproduction. Finally, we discuss the therapeutic potential of NAD^+^ augmentation through NAD^+^ precursors and CD38 inhibition, particularly emphasizing their impact on female reproductive aging and fertility. By unraveling the complex interplay between NAD^+^ biology and female reproductive aging, this review aims to establish a comprehensive foundation for future research and interventions geared toward preserving and enhancing female reproductive health.

## 2. Factors Contributing to Ovarian Aging

The ovary plays a central role in mammalian reproduction and serves two primary functions: the production of oocytes and the secretion of hormones [19]. The follicles, which serve as the fundamental functional units within the ovary, consist of a germ cell (oocyte) surrounded by somatic cells (granulosa cells and theca cells). The mammalian ovary has two distinct follicle pools: resting or non-growing follicles (NGFs) and actively growing follicles [20]. A primordial resting follicle consists of an oocyte surrounded by a single layer of flattened pre-granulosa cells. It is arrested in the diplotene/dictyate stage of meiotic prophase I [21,22]. These follicles remain quiescent throughout an individual’s life unless they experience atresia or are recruited for maturation [23]. NGFs constitute the OR, representing the functional capacity of the ovary, which in turn signifies the quantity and quality of oocytes it contains [20]. The development of the primordial follicle pool is a complex and regulated process. In most mammalian species, the maximum number of primordial follicles is determined during pregnancy or shortly after birth [24]. In human females, the peak of ovarian follicular count is achieved during intrauterine development around the 20th week of gestation, with a recorded count of about 6–7 million follicles. During the second half of fetal life, follicles undergo apoptosis, leaving this number at only 1–2 million and 300,000–400,000 at birth and menarche, respectively [25,26,27]. The number of follicles that reach the ovulation stage throughout a woman’s reproductive lifespan is approximately 400–500, one percent of the total follicular pool [28,29].

A diminishing ovarian follicle pool dictates critical reproductive processes, such as a decline in fecundity, infertility, irregular menstrual cycles, and ultimately menopause around the age of 51, with only about 1000 follicles remaining in the ovaries [25,26,27]. The chapter of the ovarian lifespan concludes when the once-robust pool of ovarian reserve diminishes to a critical threshold with advanced age. Another hourglass commences its rotation as the ovaries age, representing the post-reproductive lifespan and indicating the remaining days of an individual’s life. The genetic integrity of oocytes declines with age, leading to a cessation of natural fertility around a decade before menopause. Menopause serves as an irreversible threshold, propelling females towards an accelerated path of aging (disease accumulation), while premature menopause carries even more severe consequences [30,31]. The loss of sex hormones produced by the ovarian follicles can disrupt many health preservation mechanisms, as these hormones are directly involved in maintaining the internal homeostasis of the organism [32]. These altered homeostasis balances are subsequently accompanied by alterations in fundamental aging processes at the cellular, tissue, organ, and system levels [33,34]. These changes are interconnected with the hallmarks of aging.

Aging is a natural process that impacts ovarian tissue, similar to its effects on other organs and systems within the body. Age-related changes in ovarian function are the main contributing factor to female reproductive aging. The availability of high-quality oocytes is essential for successful fertilization and embryonic development, as they form the fundamental basis for the initiation of life. Female reproductive aging in mammals is marked by a significant decline in the number and quality of follicles and oocytes [35]. The mechanisms underlying the decline of follicular pool and oocyte quality are not fully understood, but progress has been made in elucidating the metabolic, endocrine, paracrine, and genetic factors involved [36]. Hormonal changes during reproductive aging can increase the risk of age-related diseases in women. These changes can affect other body systems, indicating an increased risk for these diseases in women in the late stages of their reproductive lives [34]. These age-related diseases include diabetes, cardiovascular disease, Alzheimer’s disease, osteoporosis, urogenital atrophy, and obesity, all of which significantly impact the quality of life for aged females [37,38]. The aging process is complex and multifaceted, involving various systemic effects primarily instigated by critical cellular alterations. Within the aging research community, these well-documented cellular alterations, consisting of twelve key factors, are collectively known as the “hallmarks of aging” (Figure 1A). It is known that these cellular traits—including cellular senescence, mitochondrial dysfunction, stem cell exhaustion, telomere attrition, loss of proteostasis, epigenetic changes, altered intercellular communication, genomic instability, inflammation, deregulated nutrient sensing, disabled macroautophagy, and dysbiosis—are at the heart of the series of events that lead to age-related decline at the systemic level [39].

## 3. Hallmarks of Aging and Ovarian Function

In the context of female reproduction, ovarian aging is a specific aspect influenced by molecular mechanisms intertwined with broader aging hallmarks (Figure 1A,B). Aging is attributed to progressive cellular damage accumulation over time. Mammalian oocytes are highly susceptible to oxidative and DNA damage due to extended prophase arrest, with the dysfunctional DNA damage response (DDR) contributing to ovarian aging [40]. The primary cause of this susceptibility is the accumulation of spontaneous damage to the mitochondria arising from increased reactive oxygen species (ROS) in oocytes, generated by the mitochondria themselves during daily biological metabolism. Mitochondrial dysfunction reduces ATP synthesis and influences the meiotic spindle assembly responsible for chromosomal segregation. Mitochondrial dysfunction, characterized by decreased mitochondrial DNA (mtDNA) copy number and function, contributes to poor oocyte quality, diminished ovarian reserve, and infertility. Proposed mechanisms include mtDNA mutations, altered metabolism, and oxidative damage. The oxidative stress resulting from age-associated mitochondrial dysfunction and ROS production can also accelerate telomere shortening by generating irreparable single-strand breaks in telomere regions. Shorter telomeres and lower telomerase activity in granulosa cells have been associated with premature ovarian failure [41]. Studies have also suggested that oxidative stress, a significant factor in aging, can lead to the accumulation of damaged proteins, further challenging proteostasis within ovarian cells. The maintenance of proteostasis is crucial for oocyte quality and ovarian reserve [42]. For instance, the gene *Clpp*, which is involved in mitochondrial protein quality control, has been found to be associated with oocyte quality and the ovarian aging phenotype in mice [43]. 

Several studies have shown that inflammation, particularly “inflammaging,” is associated with ovarian aging and can induce lesions, contributing to premature ovarian insufficiency (POI) [44,45]. The age-related decrease in primordial, primary, secondary, and antral follicles coincides with an increase in the intra-ovarian and systemic expression levels of major pro-inflammatory and inflammasome-related genes and proteins [46]. Inflammatory processes are linked to mitochondria, and several factors, including oxidative stress, cytokines, cell death signaling, and autophagy, are involved in inflammation-related ovarian aging [47]. Inflammation-associated molecular patterns promote cellular senescence, which can lead to further inflammation, creating a vicious cycle. Research has shown that the accumulation of senescent cells and the upregulation of senescence-related markers precede the decline in primordial follicle reserve, indicating a potential role for cellular senescence in ovarian aging. While the exact mechanisms and consequences of cellular senescence in ovarian aging are still being elucidated, the removal of senescent cells has been proposed as a potential therapeutic approach to preserve ovarian function [10]. Additionally, impaired autophagy may contribute to the accumulation of damaged organelles and proteins in ovarian cells, further exacerbating cellular senescence and the decline in ovarian function. 

Several studies have highlighted the significance of epigenetic modifications, such as changes in DNA methylation levels, histone modifications, and non-coding RNA expression, in the aging process of germ cells and early embryos [48,49]. Abnormal DNA methylation on critical gene promoters in granulosa cells and cumulus cells has been associated with adverse effects on ovarian function in older women [50]. Altered intercellular communication is another hallmark of aging. Intercellular communication between the oocyte and its associated somatic cells is essential for determining oocyte quality and ovulation. This communication involves various mechanisms, such as the transmission of paracrine factors, signaling molecules, and metabolites [51]. Additionally, intercellular communication helps maintain adequate oocyte ATP levels and antioxidant metabolite supply, which are important for oocyte quality. The decline in intercellular communication due to factors such as reproductive aging and in vitro aging can lead to a decrease in oocyte competence. Therefore, understanding and preserving intercellular communication is critical for maintaining female fertility, especially as women age [52].

Gut microbiota dysbiosis has been associated with polycystic ovary syndrome (PCOS), impacting metabolic and endocrine functions, and potentially affecting ovarian function through androgen-induced dysbiosis. Several studies have demonstrated the role of gut microbiota in the occurrence and development of PCOS, with implications for the regulation of insulin synthesis and secretion, and the modulation of androgen metabolism and follicle development [53,54,55]. PCOS is a complex endocrine disorder characterized by ovarian dysfunction, hyperandrogenism, and metabolic disturbances. Research has shown that PCOS is associated with decreased NAD^+^ levels in muscle tissue, which correlates with the distinct pattern of insulin resistance observed in PCOS patients. Given that NAD+ plays a crucial role in maintaining cellular energy and redox homeostasis, the depletion of NAD^+^ observed in PCOS may contribute to the accelerated ovarian aging seen in this condition. 

## 4. NAD^+^ and Its Metabolism

### 4.1. Overview of NAD^+^ and Its Importance in Cellular Functions

Nicotinamide adenine dinucleotide (NAD^+^) is a ubiquitous coenzyme involved in cellular energy production and regulating various cellular processes. NAD^+^ is crucial in multiple metabolic processes, such as glycolysis, oxidative phosphorylation, pyruvate dehydrogenase complex, and the tricarboxylic acid cycle [56]. NAD^+^ is required for over 500 enzymatic reactions [57,58] and plays a crucial role in cell signaling, survival pathways, and various cellular functions such as metabolic pathways, DNA repair, chromatin remodeling, endocrine signaling, cellular senescence, inflammation, apoptosis, proliferation, mitochondrial function, lipid and glucose homeostasis, and immune cell function [57]. The involvement of NAD^+^ in the physiological process of aging has gained attention in recent years. NAD^+^ serves as a necessary cofactor for non-redox NAD^+^-dependent enzymes, including sirtuins (SIRTs), CD38, and poly(ADP-ribose) polymerases (PARPs) [18,59,60]. The involvement of these enzymes in key cellular processes and functions is critical for maintaining tissue and metabolic homeostasis and the aging process. 

It has been established that NAD^+^ levels in tissues and cells gradually decline with age in various model organisms, including humans and rodents. Decreases in NAD^+^ levels are associated with hallmarks of aging. Recent research investigations have demonstrated that age-related conditions, such as cancer, metabolic disorders, Alzheimer’s disease, neurodegenerative diseases, and fertility decline, are linked to alterations in NAD^+^ homeostasis [56,61,62]. Many cellular and systemic processes depend on NAD^+^ homeostasis, which balances NAD^+^ synthesis and consumption. NAD^+^ biosynthesis and degradation pathways are crucial to understanding NAD^+^’s role in aging (Figure 1C).

### 4.2. NAD^+^ Biosynthesis Pathways

Cells require adequate NAD^+^ levels to fulfill both redox and non-redox NAD^+^ needs. When utilized as a coenzyme, NAD^+^ levels remain stable, but during non-redox reactions, they diminish from the cellular pool, demanding a continuous dinucleotide synthesis [63]. Other reviews have covered the details of NAD^+^ synthesis pathways [13,63,64,65]. In this section, we will briefly discuss how these pathways relate to aging and female reproduction.

#### 4.2.1. De Novo Biosynthesis Pathway

Figure 2A illustrates the de novo biosynthesis pathway of NAD+ in mammalian cells. In mammalian cells, tryptophan (Trp) plays an important role in the de novo biosynthesis pathway of NAD^+^. The major catabolic pathway of Trp is the kynurenine pathway (KP), which ultimately leads to the biosynthesis of NAD^+^. The de novo NAD^+^ biosynthesis pathway in humans comprises both enzyme- and non-enzyme-based reactions. The process of NAD^+^ synthesis starts with the amino acid Trp. Tryptophan 2,3-dioxygenase (TDO) converts Trp into quinolinic acid (QA) [65]. Another enzyme called quinolinate phosphoribosyltransferase (QPRT) converts QA to NaMN. NaMN is then converted to NAAD with the help of nicotinamide mononucleotide adenylyltransferase (NMNAT). NAD synthase (NADS) then converts this NAAD to NAD^+^ [13,66]. The de novo pathway is less important for NAD^+^ synthesis in mammals but is the primary pathway in some bacteria and yeast.

The dysregulation of the kynurenine pathway may have implications for developing pathological pregnancies [67,68]. According to recent studies, women with polycystic ovary syndrome (PCOS) have a dysregulated tryptophan–kynurenine pathway and significantly elevated levels of tryptophan [69]. 

#### 4.2.2. Preiss–Handler Pathway (PHP)

Nicotinic acid (NA) is used as a precursor in the production of NAD^+^ via the PHP route. The amino acid NA is a precursor to the coenzyme NAD^+^ and can be found in a wide variety of foods and dietary supplements. As shown in Figure 2A, the enzyme known as nicotinic acid phosphoribosyltransferase (NAPRT) is responsible for facilitating the conversion of nicotinic acid (NA) to nicotinic acid mononucleotide (NAMN). In the final two steps of the PHP pathway, NAD^+^ is produced sequentially from nicotinic acid adenine dinucleotide (NAAD) via NMNAT 1/2/3 and NADS. NMNATs are required for both the Preiss–Handler pathway and the salvage pathway, and they play critical roles in the development of mammalian embryos and the function of the female reproductive system.

#### 4.2.3. Salvage Pathway

The salvage process is comparatively more efficient for NAD^+^ synthesis in mammals than in the de novo system. The precursors utilized in this process are NAM, NA, NR, and NMN, which are derived from the recycling of NAD^+^ through consumption processes. Nicotinamide phosphoribosyltransferase (NAMPT) is the enzyme that acts as the rate-limiting step in the salvage route by catalyzing the conversion of nicotinamide (NAM) to NMN. NMN undergoes enzymatic conversion by NMNAT enzymes to produce NAD^+^. Additionally, the precursor NR is also utilized in the salvage pathway. Nucleoside transporters facilitate its transportation into cells, followed by its phosphorylation through the action of NR kinases 1 and 2 (NRKs), resulting in the production of NMN. The NMNAT enzymes facilitate the enzymatic conversion of NMN into NAD^+^. The salvage pathway is thought to be the dominant source of NAD^+^ biosynthesis. A significant differentiation among these biosynthetic processes lies in the fact that the de novo and Preiss–Handler pathways are dependent on amino acids derived from the diet and precursors of vitamin B3. In contrast, the salvage pathway can employ precursors originating from within the cell.

### 4.3. NAD^+^-Consuming Pathways

#### 4.3.1. NAD^+^ Consumption by Sirtuins and Their Role in Female Reproduction

The sirtuin (SIRT) protein family belongs to class-III histone deacetylases (HDACs) and are evolutionarily conserved proteins that utilize NAD^+^ as a co-substrate. In mammals, the sirtuin family comprises seven genes and proteins (SIRT1–SIRT7), characterized by a discrete pattern of subcellular localization, target molecules, and effects. The sirtuins that are predominantly localized in the nucleus include SIRT1, SIRT6, and SIRT7, and these play an important role in the regulation of gene transcription. SIRT3–SIRT5 have mitochondria-targeting sequences, are primarily localized in the mitochondria, and play important roles in regulating energy metabolism, mitochondrial function, and cellular homeostasis. SIRT2 is predominantly localized to the cytoplasm and targets and regulates protein expression. The subcellular localization of sirtuins has been a topic of debate and research. While there is evidence that some sirtuins shuttle in and out of their primary target site, the exact mechanisms and extent of this shuttling are still being investigated. For example, SIRT1 shuttles to the cytoplasm under specific circumstances, whereas SIRT2 relocates in the nucleus when the cell cycle transitions from the G2 to M phase [70,71]. 

Since their discovery in mammals, SIRTs have emerged as critical regulators in various cellular processes, which include cell survival, histone modification, transcription regulation, chromatin remodeling, genome stability, energy metabolism modulation, inflammation, regulation of ion channels, and modulation of the cellular redox state [72,73,74,75,76]. In the context of female reproduction, SIRTs have been linked to various functions, encompassing folliculogenesis, embryogenesis, fertility regulation, and protection against oxidative stress [77]. Most of these investigations have been conducted using animal models, but there is an increasing interest in research involving human subjects. Genetically modified mice have shown that certain genes interact with sirtuin signaling to regulate the primordial follicle pool dynamics. Subsequently, various transgenic models have expanded our understanding of sirtuins in reproductive functions, highlighting their importance in gonadal processes. Importantly, the expression of SIRTs has been detected in mammalian ovaries, oocytes, granulosa cells, and embryos, as illustrated in the comprehensive table of pertinent findings (Table 1) [78,79]. Expression of all sirtuin family members is evident in mature oocytes at the metaphase II stage of meiosis, with a subsequent decline in protein levels following the initial embryonic cleavage. This intriguing pattern suggests that SIRTs are stored during the process of oogenesis [80,81] and underscores their pivotal role in the early stages of embryonic development. 

Sirtuins are now recognized as essential regulators of key processes in oogenesis, oocyte maturation, and embryonic development [82]. Studies have shown that reduced NAD^+^ levels in oocytes from older mice can impair sirtuin activity, as sirtuins depend on NAD^+^ for their enzymatic functions, leading to a loss of oocyte quality with age, which is the rate-limiting factor for fertility [61,83]. Redox perturbations linked to aging, diseases, and metabolic dysfunctions can significantly affect fertility by disturbing the relationship between energy metabolism and female reproduction [84]. Therefore, NAD^+^ repletion and overexpression of sirtuins have been shown to rescue female fertility during reproductive aging [61]. Moreover, a rise in NAD^+^ levels strongly correlates with sirtuin activation during fasting and caloric restriction [56,85]. 

SIRT-1: SIRT1, an extensively researched sirtuin, is essential for histone deacetylation and the regulation of transcription factors. Several factors have been identified to play a role in this process, including NFκB, p53, PARP1, PGC1α, and FOXOs [74,86,87,88,89]. The importance of sirtuins in the control of fertility was first brought to light in 2003 when it was discovered that mice lacking SIRT1 had impaired reproductive functions [90]. When transgenic mice overexpressing SIRT1 exhibited a delay in reaching sexual maturity, the possibility of sirtuins impacting folliculogenesis emerged [91]. In 2016, Cinco et al. suggested that SIRT1 plays a role in follicle NAD^+^ metabolic changes during follicular growth that initiates from the primordial follicle [92]. This study demonstrated that SIRT1 nuclear expression in oocytes increased during the awakening of primordial follicles, accompanied by a reduction in the nuclear NADH/NAD^+^ ratio. This phenomenon is required in oocytes to transition from glycolytic to oxidative phosphorylation metabolism to facilitate growth. This perspective supported the notion that SIRT1 has the potential to stimulate PGC1α, thereby promoting mitochondrial biogenesis and oxidative phosphorylation. Based on these findings, it is possible to consider sirtuins as energy sensors that function as paracrine factors in the ovary. They also serve as internal regulators of oocyte development, influencing primordial follicles’ fate. Moreover, it has been observed that porcine granulosa cells (GCs) transfected with SIRT1 show an elevated expression of markers associated with cell proliferation. This finding implies that SIRT1 may play a role in the final differentiation of granulosa cells during the process of oocyte maturation [93].

SIRT-2: SIRT2 is a critical regulator of carbohydrate metabolism and cell division. It blocks the proteasomal degradation of PEPCK1, the key enzyme in gluconeogenesis, and deacetylates α-tubulin and BubR1, which are essential for precise chromosome segregation and microtubule–kinetochore interactions in mitosis [94,95,96,97]. SIRT2 is also required for oocyte maturation and meiosis. It is evident that an SIRT2 inhibitor blocked GVBD progression in mouse oocytes in vitro, and SIRT2 knockdown disrupted spindle organization and chromosome alignment in meiosis [98,99]. In mouse oocytes, SIRT2 levels decrease with age, contributing to age-dependent spindle defects and chromosome disorganization. Overexpression of SIRT2 in aged oocytes decreases acetylation of H4K16, α-tubulin, and BubR1, reducing the occurrence of meiotic defects associated with maternal age [99,100]. Given SIRT2’s role in spindle stability, it is hypothesized that decreased SIRT2 levels may play a role in postovulatory oocyte aging, potentially influenced by changes in tubulin acetylation.

SIRT-3: As described earlier, SIRT3–SIRT5 with mitochondria-targeting sequences are primarily localized in the mitochondria. SIRT3, the most investigated mitochondrial sirtuin, significantly impacts key mitochondrial functions, encompassing energy production, ketogenesis, regulation of ROS, β-oxidation, and cell death [101]. Mitochondria serve as central hubs for multiple metabolic pathways. Dysfunctions in these organelles can disrupt redox balance, increasing the risk of aging and metabolic disturbances [102]. Refs. [103,104] demonstrated that overexpression of SIRT3 in mice leads to efficient protection against ROS by deacetylating SOD2, offering protection against oxidative stress and age-related pathologies. Conversely, SIRT3^−/−^ mouse embryonic fibroblasts exhibit overproduction of ROS, reduced chromosomal stability in response to external stress, and decreased oxidative phosphorylation [105]. Furthermore, in mouse obesity models, reduced SIRT3 expression correlates with elevated ROS levels, while SIRT3 overexpression mitigates ROS production, reducing spindle defects and chromosome misalignment [106]. Similarly, impaired SIRT3 expression in human oocytes leads to decreased mitochondrial biogenesis, compromising developmental competence. Interestingly, a retrospective analysis conducted by the same study revealed a reduction in embryo quality among patients undergoing in vitro maturation (IVM) and a higher rate of spontaneous abortions in contrast to those in ovarian stimulation cycles [107]. SIRT3 collaborates with SIRT1 to regulate mRNA expression of aromatase, cholesterol side-chain cleavage enzyme (CYP11A1), steroidogenic acute regulatory protein (StAR), 17β-hydroxysteroid dehydrogenase 1 (17β-HSD1), and 3β-hydroxysteroid dehydrogenase (3β-HSD) in human GCs. SIRT3 positively impacts folliculogenesis and luteinization through its regulation of steroidogenesis, progesterone secretion (P4), and reduction of ROS-induced stress [108]. Furthermore, SIRT3 functions as a metabolic sensor in both GCs and CCs by regulating necessary mitochondrial enzymes such as glutamate dehydrogenase (GDH). Decreased SIRT3 activity and increased acetylated GDH levels are characteristic features of aging in GCs and CCs. Hence, changes in mitochondrial sirtuins could potentially impact post-translational modifications of mitochondrial proteins, leading to metabolic shifts within aging follicles [109]. Ref. [110] demonstrated that high palmitic acid-induced inhibition of SIRT3 expression leads to the AMPK/SIRT3 pathway’s downregulation, culminating in ceramide accumulation, elevated acetylation of mitochondrial proteins, and dysfunction in porcine oocytes. 

SIRT-4: SIRT4 is the second mitochondrial sirtuin protein and mediates ADP-ribosylation of glutamate dehydrogenase, thus decreasing carbon flux in the Krebs cycle [111,112]. Its significance in energy metabolism has been further established through its lipoamidase function, which modulates pyruvate dehydrogenase complex (PDH) activity, thereby impacting acetyl-CoA production [113]. Regarding SIRT4, only a limited number of studies have been conducted to ascertain its impact on female fertility. SIRT4-knockout models, developed to explore SIRT4’s function in different tissues and organs, were fertile [111,114,115]. Nonetheless, it is worth mentioning that Sirt4 expression is initially elevated in mature oocytes and zygotes but decreases in two-cell embryos. Sirt4 transcripts stay consistently low during cleavage and eventually become undetectable by the time of the blastocyst stage [80]. In contrast to SIRT3, SIRT4 overexpression adversely affects oocyte quality and competence, leading to problems such as impaired meiosis, disrupted MII spindle formation, and changes in mitochondrial distribution [114]. Higher SIRT4 expression is also associated with reduced ATP production and elevated ROS levels, particularly in oocytes undergoing postovulatory aging [116].

SIRT-5: SIRT5, the third mitochondrial sirtuin, possesses deglutarylase, desuccinylase, and demalonylase activities [117,118]. Similar to SIRT4, SIRT5’s role in female reproductive potential remains largely unexplored, with only a few studies conducted. Expression of Sirt5 has been identified in mouse oocytes and preimplantation embryos, with higher levels in mature oocytes and zygotes and lower levels in embryos from the two-cell to the blastocyst stages [102]. In human GCs, two studies have focused on SIRT5. Pacella-Ince and colleagues (2014) confirmed the presence of Sirt5 transcripts, proteins, and activity, noting reduced SIRT5 in cells from women with advanced reproductive age [109,119]. In contrast, Gonzales-Fernandez et al. (2019) observed higher SIRT5 expression in patients over 40 years old and a positive correlation between Sirt5 expression and the number of oocytes collected from poor-responder patients [120]. Understanding the specific contributions of other mitochondrial sirtuins (SIRT4 and SIRT5) to female reproduction remains an area ripe for exploration. Comprehensive research could uncover their involvement in key processes like oocyte maturation, embryo development, and hormonal regulation, shedding light on their relevance in fertility. It is imperative that future studies delve deeper into the functions of SIRT4 and SIRT5 in female reproduction to provide a more comprehensive understanding of mitochondrial sirtuin involvement in fertility-related processes. In doing so, we may unravel new therapeutic targets and strategies to enhance reproductive outcomes and address fertility challenges in females.

SIRT-6: SIRT6 has recently been identified as a regulator of female reproductive function, describing that it possesses deacetylase and ADP-ribosyltransferase activity [121]. The downregulation of SIRT6 using siRNA during the in vitro maturation (IVM) of mouse oocytes has been observed to have several effects. These include a reduction in meiotic development, an increase in aneuploid MII eggs, and the presence of larger spindles, chromosome misalignment, and aberrant K-MT attachments during metaphase I [122]. Embryos derived from these metaphase II eggs through in vitro fertilization (IVF) display reduced efficacy in progressing to the blastocyst stage. The protein SIRT6 has been found to be linked to chromatin throughout the process of oocyte maturation, whereas it has distinct characteristics compared to its function in somatic cells. The study involving SIRT6-depleted oocytes showed no changes in H3K9ac and H3K56ac, but an increase in H4K16ac was observed. This implies that the epigenetic targets of SIRT6 may exhibit variability across different cell types. The importance of investigating the impact of SIRT6-dependent H3K18ac regulation in meiotic cells, specifically in the establishment of amphitelic K-MT attachments, arises from its known role in mitosis in somatic cells [121].

SIRT-7: The SIRT7 protein is predominantly concentrated in the nucleolar region of actively proliferating cells. Its function involves the regulation of ribosomal DNA (rDNA) transcription [123,124]. The relocation of SIRT7 from nucleolar regions to the cytosol is connected to a decline in rDNA transcription, which is a defining feature of replicative senescence [125]. SIRT7 plays a crucial role in female reproduction and reproductive longevity in mice [126]. SIRT7 siRNA-mediated downregulation results in reduced primordial follicles and early infertility due to defective chromosome synapsis during meiotic prophase [127]. Although SIRT7 is known as a histone H3K18 deacetylase in somatic cells, its role in meiotic chromosome synapsis does not involve H3K18 deacetylation [128]. The exact mechanism of SIRT7’s involvement in chromosome synapsis remains to be determined. In oocyte meiotic maturation, SIRT7 depletion leads to aneuploidy, delayed development, spindle abnormalities, polar body issues, and mitochondrial dysfunction. These effects can be mitigated by antioxidant treatment, emphasizing SIRT7’s role in redox homeostasis during meiosis and early development. Notably, Sirt7−/− mice have a milder phenotype than SIRT7-depleted oocytes, possibly due to phenotypic penetrance differences or developmental compensation. Overall, SIRT7 emerges as a crucial player in maintaining the ovarian reserve, ensuring proper chromosome synapsis during meiosis and preserving reproductive function. Its diverse roles in oocyte meiotic maturation and the regulation of retrotransposons underscore its significance in female reproduction, calling for further exploration of its mechanisms and potential therapeutic implications. Further investigations are needed to unravel the exact molecular mechanisms modulated by SIRT7 during female reproduction.

#### 4.3.2. NAD^+^ Consumption by PARPs and Their Role in Female Reproduction

Poly(ADP-ribose) polymerases (PARPs), sometimes referred to as NAD⁺ ADP-ribosyltransferase 1 or poly[ADP-ribose] synthase, constitute a substantial family of around 17 enzymes (PARP1–17) [129]. These enzymes are primarily involved in robust mechanisms for both the detection and repair of DNA damage response (DDR). Out of all the poly(ADP-ribose) polymerases (PARPs), PARP1, PARP2, and PARP3 are the only ones that exhibit nuclear localization in response to early DNA damage, as depicted in Figure 1C and Figure 2B. These particular PARPs play a crucial role in the repair of DNA damage, as well as in processes such as apoptosis, regulation of chromatin structure, and the cell cycle checkpoint [130,131]. 

PARP1, being the most widely researched member within this family, plays a significant role in nearly 90% of the overall PARP activity, particularly in relation to DNA damage response [132,133]. Upon encountering DNA strand breaks, the DNA-binding domain of PARP1 identifies the locations of damage and then forms a binding interaction with them. Following this, the catalytic domain of the enzyme begins the formation of poly(ADP-ribose) chains, triggering the activation of the DNA repair machinery. This particular process is commonly known as PARylation. Poly(ADP-ribose) polymerase (PARP) enzymes catalyze the hydrolysis of nicotinamide adenine dinucleotide (NAD^+^), resulting in the generation of nicotinamide (NAM) and ADP-ribose as by-products. Elevated PARP1 activity results in significant depletion of NAD^+^ levels in the context of DNA damage. The correlation between PARP1 and the aging phenomenon is mostly ascribed to its function as a signaling molecule that responds to NAD^+^. Nevertheless, it is crucial to acknowledge that PARP1 substantially impacts the reduction in NAD^+^ levels, not only in cells that have been subjected to sudden DNA damage but also in normal physiological states and a range of pathological conditions [134]. This highlights the key significance of PARP1 in the regulation of NAD^+^ equilibrium. In the context of female reproduction and reproductive aging, research on PARP is limited, yet there is a growing interest in exploring its role. The roles of PARPs have been investigated in reproduction, aging, stem cells, inflammation, metabolism, and particularly cancer [135,136]. Some studies have found that PARP1 plays an important role in ensuring the stability of chromosomes during critical stages of meiosis within the female germ line. Oocytes deficient in PARP1 exhibit incomplete homologous synapses and prolonged phosphorylation of histone H2A.X. This condition makes the female gamete susceptible to genome instability [137]. Interestingly, PARP dysfunction paradoxically contributes to the development of immune-related ovarian failure. 

PARP1 is essential for the activation of NF-κB, a crucial signaling pathway for immune responses, in response to ionizing radiation [138]. Inhibition of PARP1 activity during in vitro maturation of pig oocytes resulted in suppressed cumulus expansion, impaired embryo development, decreased total number of blastocyst cells, and increased apoptosis [139]. In mice, inhibition of PARP resulted in developmental arrest and interfered with spindle formation and tubulin polymerization [140]. Moreover, the inhibition of PARPs during in vitro embryo culture leads to enhanced pronuclear formation, increased embryo fragmentation, and hindered blastocyst formation. These effects are likely attributed to the improper localization of ADP-ribose polymers [141]. The inhibition or genetic deletion of PARP1 and PARP2 in mice led to pregnancy loss caused by impaired decidualization. This impairment was characterized by elevated p53 signaling and the presence of senescent decidual cells [142]. 

As already discussed, NAD^+^ levels negatively correlate with increasing age. Conversely, PARP activity significantly increases with age and inversely correlates with tissue NAD^+^ levels, reducing efficiency in repairing DNA damage in oocytes. This can result in an accumulation of DNA mutations and chromosomal abnormalities, increasing the risk of infertility and diminishing the ovarian reserve, ultimately leading to ovarian aging. Recent studies have shown that PARP inhibitors, such as olaparib, can deplete the ovarian reserve and primordial follicles in mice, which raises concerns about the potential impact of PARP inhibitors on female fertility in humans [143]. PARP inhibitors have also been shown to cause ovarian toxicity and reduce ovarian reserve in mice [144]. These findings suggest that age-related changes in PARP activity and NAD^+^ availability may contribute to age-related decline in fertility. Of note, PARP inhibitors (PARPi) have been developed primarily for cancer therapy, but their mechanism of action presents a paradoxical role in fertility. Further research is needed to fully understand the potential impact of PARP modulation on female reproduction. 

#### 4.3.3. NAD^+^ Consumption by CD38 and Its Role in Female Reproduction

CD38, a multifunctional protein that functions as both a receptor and an enzyme, plays an essential role in regulating cellular and tissue NAD^+^ levels. It is encoded by homologous genes located on chromosomes 4 and 5 in humans and mice, respectively. CD38 is predominantly found on the cell surface as an ectoenzyme, with its catalytic site facing the extracellular space [145]. The localization of CD38 NADase at the extracellular compartment, while the majority of NAD^+^ is found intracellularly, presents a significant topological paradox in NAD^+^ biology and metabolism. Until recently, elucidating this paradox posed a challenge. CD38 has been observed in various intracellular organelles, such as the mitochondria, nucleus, and endoplasmic reticulum. Additionally, both soluble intra- and extracellular forms of CD38 have been observed [146,147]. Recent research has demonstrated that CD38 degrades NAD^+^ as well as circulating NAD^+^ precursors [148]. Understanding the distinct roles played by CD38 in maintaining NAD^+^ levels within different “anatomical” locations of cells and tissues remains one of the outstanding questions in this research area.

CD38 plays an essential role in various cellular physiological processes, including the regulation of immune cell function, nuclear Ca^2+^ homeostasis, inflammation, and metabolic processes such as glucose and lipid homeostasis [149,150]. CD38 has been detected in various cells and tissues, including immune cells, the liver, testis, kidney, and brain. Recently, its expression in the ovary has also been documented [62]. It is crucial for intracellular calcium signaling and generates the second messengers adenosine diphosphoribose (ADPR) (NAD^+^-glycohydrolase activity) and cyclic ADPR (cADPR) (cyclase activity) [151,152]. Additionally, it exhibits hydrolytic activity, converting cADPR to ADPR. Furthermore, under acidic pH conditions, it functions as a nicotinic acid adenine dinucleotide phosphate (NAADP) synthase, converting NADP1 to NAADP in the presence of nicotinic acid. It also degrades NAADP into ADPR via its NAADP-hydrolase activity. All of these reaction products are second messengers involved in the regulation of cytoplasmic Ca^2+^ fluxes [151], and the full spectrum of their biological functions is being actively elucidated. In the context of female reproduction, intracellular calcium signaling plays a critical role in folliculogenesis, ovulation, fertilization, and egg activation. The role of intracellular Ca^2+^ signaling in folliculogenesis is supported by the observation that FSH increases intracellular Ca^2+^ in granulosa cells [153].

CD38 deficiency in mice resulted in an increased number of follicles and enhanced fertility [62]. These findings suggested that CD38 might have played a role in inhibiting follicle development, apoptosis, and follicular atresia. It was hypothesized that CD38 influenced granulosa cell functions during follicular development through its involvement in calcium signaling. This might have also had implications for the selection and maturation of dominant follicles during the menstrual or estrous cycles. Furthermore, intracellular Ca^2+^ release channels and Ca^2+^-selective plasma membrane channels were proposed to have played important roles in ovulation and fertilization [154]. Oocytes of higher quality were more likely to have been fertilized and developed into healthy embryos. Studies demonstrated that CD38 deficiency in mice led to improved oocyte quality [62]. CD38 has been shown to play a crucial role in regulating important metabolic processes, including glucose tolerance and insulin sensitivity, which could have significantly influenced the reproductive system [148,155,156]. CD38 was shown to have impacted hormonal regulation, specifically the release of oxytocin. Additionally, CD38 was regulated by estrogen and progesterone, both of which were steroid hormones [157,158].

It is evident that CD38 plays a crucial role in the decline in NAD^+^ levels in mammals as they age, which is influenced by its ectoenzyme or endoenzyme activity. An age-related increase in the activity of CD38 has been described in the liver, spleen, adipose tissue, skeletal muscle [148], kidney, heart [159,160], brain [161], and ovary [62]. CD38-mediated NAD^+^ depletion has been linked to a number of age-related diseases, including cardiovascular disease, neurodegenerative disease, cancer, autoimmune diseases, Alzheimer’s disease, and type 2 diabetes [162]. Similar to other body tissues, the level of CD38 increases in female ovarian tissue with age and is the cause of the decline in the level of NAD^+^ in the ovary [62]. CD38-knockout (KO) animals have been used to study the role of CD38 in aging and metabolic dysfunction. In comparison to wild-type controls, CD38-KO young mice have greater primordial follicle pools, higher ovarian NAD^+^ concentrations, and increased fecundity. The increased OR can be attributed to a prolonged window of follicle formation in the early stages of development. At an advanced age, however, the beneficial effect of CD38 loss on reproductive function is no longer maintained [62]. 

CD38 is regulated by several factors, including nuclear factor κB (NF-kB), liver X receptor (LXR), and STAT [163,164]. Significantly, certain nuclear receptors, inflammatory cytokines, endotoxins, and interferons are capable of stimulating CD38 expression [163]. Consequently, treatment with lipopolysaccharide (LPS) and TNF-a leads to an upregulation of CD38/NADase activity and a consequent significant reduction in cellular NAD^+^ concentrations. Numerous studies have determined that LPS treatment not only triggers inflammation but also reduces NAD^+^ levels within granulosa cells in women diagnosed with polycystic ovary syndrome (PCOS). Meanwhile, it has become increasingly evident that women with PCOS frequently exhibit metabolic characteristics linked to decreased NAD^+^ levels, including obesity, insulin resistance, and hepatic steatosis. Another study found that the NAD^+^ levels were reduced after activation of inflammation in a human granulosa-like tumor cell line (KGN) treated by LPS [165]. Understanding the regulatory mechanisms of CD38 gene expression can provide insights into the mechanisms that govern the elevated expression of CD38. The pro-inflammatory state observed during aging could be associated with cytokine-induced CD38 expression, age-related tissue NAD^+^ decline, and the theory of ”inflammaging” [17,166]. CD38 is being recognized as a potential target for therapeutic intervention in various physiological and pathological conditions, including aging and age-related diseases, due to its involvement in NAD^+^ homeostasis.

**Table 1 ijms-25-04680-t001:** Role of NAD^+^ and associated genes in female reproduction.

Area	Role of NAD^+^	Genes	References
Oocyte development and maturation	Energy production through mitochondria	NAMPT, SIRT1, SIRT3	[82,167,168]
	DNA repair and maintenance	PARPs (specifically PARP1)	[143,169,170]
	Cell signaling and gene expression	SIRT1, SIRT2, SIRT3	[77,79,171]
	Maintenance of cellular redox balance	SIRT3, SIRT4	[75,102,172,173,174]
Follicle development and selection	Regulation of granulosa cell proliferation and differentiation	SIRT1, SIRT3, SIRT6	[175,176,177,178,179]
	Follicle-stimulating hormone (FSH) signaling	SIRT1	[180,181]
	Estrogen biosynthesis	SIRT1, SIRT2	[182,183]
Embryo development and implantation	Mitochondrial function and ATP production in preimplantation embryos	SIRT3	[80,107]
	DNA repair and epigenetic modifications	PARP1, PARP2, SIRT1	[184,185,186]
	Regulation of cell cycle progression and differentiation	SIRT2, SIRT4, NMNAT2, CD38	[95,187,188,189]
Maternal health and pregnancy outcomes	Regulation of insulin sensitivity and glucose metabolism	SIRT1, SIRT4, NMNAT3, NAMPT, CD38	[111,148,190,191,192,193,194]
	Immune modulation and inflammation control	CD38, NAMPT	[189,195,196]

## 5. Boosting NAD^+^ as a Therapeutic Strategy for Ovarian Function

The potential therapeutic benefits of NAD^+^-boosting strategies in age-related diseases and longevity have garnered significant attention in recent years. The decline in NAD^+^ is a characteristic feature of aging and is linked to reproductive decline. Hence, addressing this decline holds the potential to enhance reproductive health and fertility in older females (Figure 3). Numerous strategies exist for increasing NAD^+^ levels. NAD^+^ levels can be increased through dietary supplementation of NAD^+^ precursors, inhibition of NAD^+^-consuming enzymes, and modulation of NAD^+^ biosynthesis enzymes. In contrast to other age-related diseases, there is relatively less research on NAD^+^-boosting therapies in the context of female reproductive aging. Hence, we will now discuss current research concerning the influence of NAD^+^ precursors on reproductive aging.

### 5.1. Supplementation of NAD^+^ Precursors

NAD^+^ precursors naturally exist in food and are used to restore NAD^+^ levels. These precursors include tryptophan, nicotinamide (NAM), nicotinic acid (NA), nicotinamide mononucleotide (NMN), and nicotinamide riboside (NR), with the latter two being more extensively studied. NAD^+^ precursors hold promise in the prevention and treatment of numerous metabolic and age-related diseases. Supplementation with NAD^+^ precursors, such as nicotinamide riboside (NR) and nicotinamide mononucleotide (NMN), can increase ovarian reserve and improve the quality of oocytes, leading to restored fertility. 

Recent research has shown that enhancing NAD^+^ levels could improve the quality of oocytes and reverse infertility in old mice [61,197]. The supplementation of nicotinic acid (NA) has been found to have a significant effect on cumulus expansion and GC proliferation in in vitro follicle culture [198]. Furthermore, adding NA to culture media for bovine oocytes and embryos during in vitro fertilization (IVF) and maturation (IVM) has improved results, such as increased blastocyst formation, improved embryo cleavage, and effective oocyte maturation with polar body formation [199,200].

NMN supplementation has been demonstrated to reverse age-related declines in egg quality and enhance fertility in mouse models. NMN is the end product in the salvage pathway of NAD^+^ synthesis. In the ovaries of mid-aged mice, NMN therapy ameliorated ovarian senescence by upregulating vital cellular processes, such as autophagy and mitochondrial function [201]. Transcript levels of Gdf9, an oocyte-secreted factor involved in oocyte developmental competence acquisition, are impacted when mice are fed a high-fat diet [202], whereas NMN supplementation was found to enhance the follicular count in these mice [202]. Additionally, in aged mice, NMN administration enhanced live birth rates and blastocyst formation rates, and restored spindle assembly [61]. Another study found that giving NMN intraperitoneally to elderly mice may improve the rates of blastocyst development and increase the rates of fertilization during IVF [203]. Conversely, a study in pigs found that adding NMN to oocyte maturation media did not improve oocyte maturation or embryonic development [204]. Species differences in energy substrate use during oocyte maturation may account for this discrepancy. In vitro, pig oocytes rely primarily on glucose, while mice predominantly use pyruvate [205,206]. Alternatively, it is possible that NMN does not have the same beneficial effects on oocyte quality and embryonic development in pigs as it does in mice. More research is needed to determine whether NMN can serve as a viable supplement for enhancing oocyte quality and embryo development in other species.

In an in vitro study, blastocyst formation rates were rejuvenated when oocytes were supplemented with NR in aged mice [61]. These findings suggest that the metabolites within the Preiss–Handler pathway and those influenced by the inhibitor have the potential to enhance specific aspects of fertility in mice. Of note, this research stands as the first of its kind to evaluate the impact of NR on oocyte quality in any species. As such, it underscores the necessity for subsequent studies to validate these findings across various contexts. However, the impact of NAD^+^ precursors on human fertility is still an evolving field of research. Although animal studies show promise, more research is needed to determine the ideal dosage for achieving the best results and to translate this research into clinical applications. 

### 5.2. CD38 Inhibitors

CD38 inhibitors have been the subject of extensive research in the context of aging and age-related diseases, with promising results in preventing NAD^+^ depletion. In the realm of female fertility, the role of CD38 inhibitors remains largely unexplored, because CD38’s role in female reproductive function was not established until recently, when researchers found that CD38 regulates ovarian function and fertility through NAD^+^ metabolism [62]. Several flavonoids, such as quercetin, luteolin, apigenin, kuromanin, and luteolinidin, have been found to inhibit CD38 activity. In animal studies, 78c has been found to have greater potency than flavonoids in reversing NAD^+^ decline during aging. This improvement in NAD^+^ levels leads to improved physiological function in several age-associated systems. However, to our knowledge, only a few studies have reported the role of 78c as a CD38 inhibitor in the context of female fertility [62,207]. By inhibiting CD38, these compounds may help maintain or restore NAD^+^ levels within the female reproductive tissues. Further studies are needed to elucidate tissue-specific CD38 functions, facilitating the development of clinical applications of CD38 inhibitors in the context of female fertility. 

### 5.3. NAD+ Biosynthesis through Nutrition

Animal-derived foods such as meat, fish, and dairy products are abundant sources of NAD^+^ precursors [208]. Plant-based foods, including cucumber, cabbage, immature soybeans, broccoli, avocado, and tomato, also contain significant amounts of NAD^+^ intermediates like nicotinamide mononucleotide (NMN) and nicotinamide riboside (NR). Broccoli contains 0.25–1.88 mg of NMN per 100 g, while avocado and tomato provide 0.26–1.60 mg/100 g [209]. Smaller quantities of NMN can also be found in raw beef, shrimp, and in human and cow milk. Human studies suggest that the daily requirement for NAD^+^ precursors is relatively small, around 15 mg of niacin (vitamin B3) equivalents, a collective term for nicotinic acid (NA) and nicotinamide (NAM) [210]. However, even modest decreases in NAD^+^ levels can have negative health consequences, so maintaining NAD^+^ homeostasis is critical. Of note, while dietary intake of NAD^+^ precursors is important, the body also has efficient internal recycling systems to maintain adequate NAD^+^ levels. This intracellular NAD+ synthesis through recycling pathways is crucial for keeping NAD^+^ levels sufficient in cells.

## 6. Conclusions and Future Directions

The decline in female fertility with age is a multifaceted process influenced by various factors, and emerging research suggests that NAD^+^ metabolism may play a crucial role in this decline. NAD^+^ is essential for cellular functions, and its levels decline with age, contributing to cellular dysfunction and the aging process. Central to cellular processes, NAD^+^ emerges as a key player in this complex interplay, influencing cellular energy production, metabolism, and survival. The hallmarks associated with aging, diseases, and metabolic dysfunctions underscore the profound impact on fertility by disrupting the delicate balance between energy metabolism and female reproduction. Enzymes such as sirtuins, PARPs, and CD38, integral to NAD^+^ biology, are identified as key players, offering potential therapeutic targets. The exploration of NAD^+^’s therapeutic potential unveils promising prospects for mitigating age-related declines in female fertility. By understanding its biosynthesis and degradation pathways, this review sets the stage for innovative interventions that may not only preserve but also enhance female reproductive health. Ultimately, the integration of NAD^+^ modulation into reproductive medicine holds promise for addressing the multifaceted challenges associated with age-related reproductive decline in women.

## Figures and Tables

**Figure 1 ijms-25-04680-f001:**
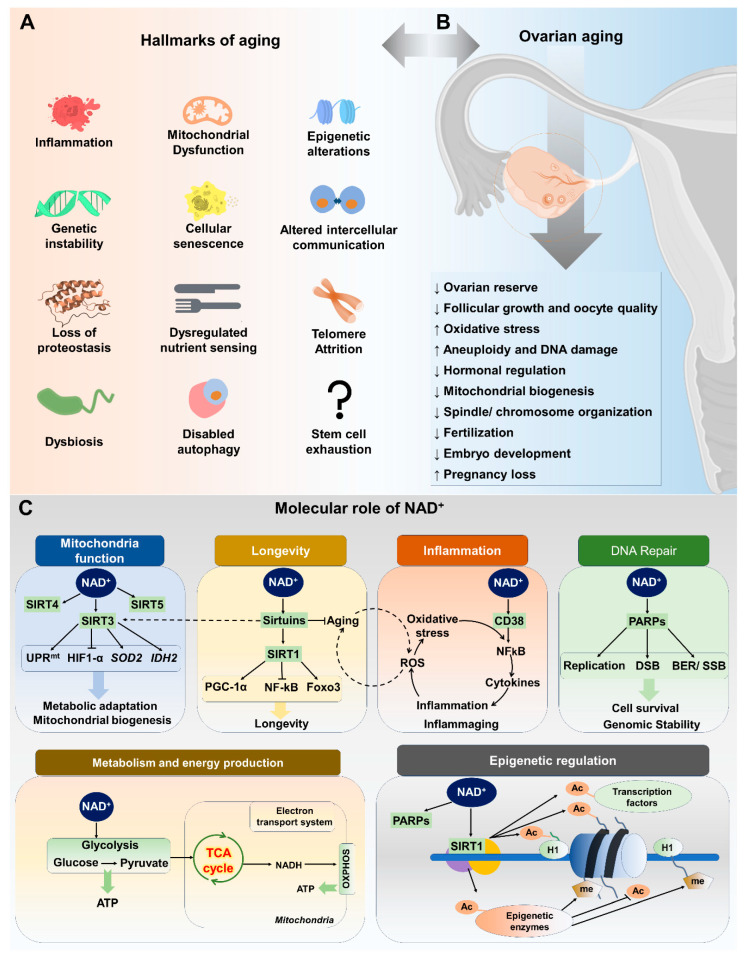
Interplay of aging hallmarks, ovarian aging, and role of NAD^+^. (**A**) Hallmarks of aging. These hallmarks of aging affect the ovarian aging process. (**B**) Ovarian aging, influenced by various hallmarks of aging, leads to a cascade of detrimental effects on ovarian reserve, follicular growth, and oocyte quality. The figure describes how increased oxidative stress, aneuploidy, and DNA damage contribute to diminished ovarian function. Moreover, disturbed hormonal regulation, decreased mitochondrial biogenesis, and compromised spindle/chromosomal organization further aggravate the decline in fertility. Ultimately, the consequences extend to decreased fertilization rates, impaired embryo development, and increased pregnancy loss. (**C**) The role of NAD^+^ in aging is very important as it plays an important role in numerous biological processes essential for cellular homeostasis and functionality. Firstly, it plays an important role in mitochondrial function through sirtuin-regulated mechanisms. Additionally, NAD^+^ is implicated in pathways associated with longevity regulation, modulating the cellular stress response and promoting longevity. It also exerts regulatory effects on inflammation, influencing immune response and inflammation through enzymes such as CD38. Furthermore, NAD^+^ serves as a substrate for enzymes such as PARPs which are involved in DNA repair pathways, contributing to cell survival and genomic stability. It participates in various metabolic pathways, including glycolysis and oxidative phosphorylation, facilitating energy production and nutrient metabolism. Moreover, NAD^+^-dependent enzymes, particularly sirtuins, regulate epigenetic modifications such as histone deacetylation and DNA methylation, thereby influencing gene expression and cellular differentiation.

**Figure 2 ijms-25-04680-f002:**
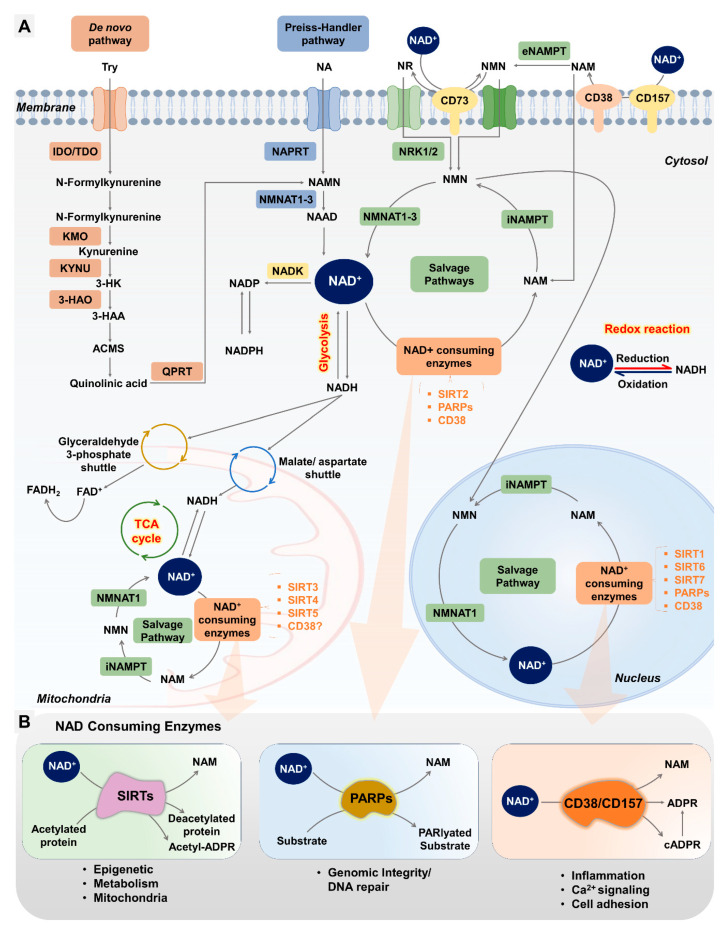
NAD^+^ metabolism (biosynthesis and consuming pathways). (**A**) NAD^+^ biosynthesis. The de novo biosynthesis pathway initiates with Trp conversion to QA by TDO. QPRT further converts QA to NAMN, ultimately resulting in NAD^+^ production via NADS. The Preiss–Handler pathway utilizes NA as a precursor, with NAPRT catalyzing NA conversion to NAMN. Subsequent steps involving NMNAT enzymes lead to NAD^+^ synthesis. The salvage pathway, more efficient in mammals, utilizes precursors like NAM, nicotinamide NR, and NMN, recycled from NAD^+^ breakdown. NAMPT catalyzes NAM to NMN conversion, while NR is phosphorylated by NRKs before conversion to NMN and then NAD^+^ via NMNAT enzymes. Notably, the salvage pathway can utilize precursors from within the cell, distinguishing it from de novo and PHP pathways reliant on dietary amino acids and vitamin B3 precursors. These pathways collectively ensure cellular NAD^+^ homeostasis critical for various physiological processes, including those vital for female reproduction. (**B**) NAD^+^-consuming enzymes. Sirtuins regulate various cellular processes including epigenetic regulation, energy metabolism, and mitochondrial biogenesis. PARPs are involved in DNA damage repair and genomic stability. CD38, acting as a NADase, influences cellular NAD^+^ levels and intracellular calcium signaling. Age-related changes in CD38 activity contribute to NAD^+^ decline, impacting the process of inflammation in the body. (Abbreviations: Trp, tryptophan; QA, quinolinic acid; TDO, tryptophan 2,3-dioxygenase; QPRT, quinolinate phosphoribosyltransferase; NaMN, nicotinic acid mononucleotide; NADS, nicotinamide adenine dinucleotide synthase; NA, nicotinic acid; NAPRT, nicotinic acid phosphoribosyltransferase; NAMN, nicotinic acid mononucleotide; NAM, nicotinamide; NR, nicotinamide riboside; NAMPT, nicotinamide phosphoribosyltransferase; NRKs, NR kinases).

**Figure 3 ijms-25-04680-f003:**
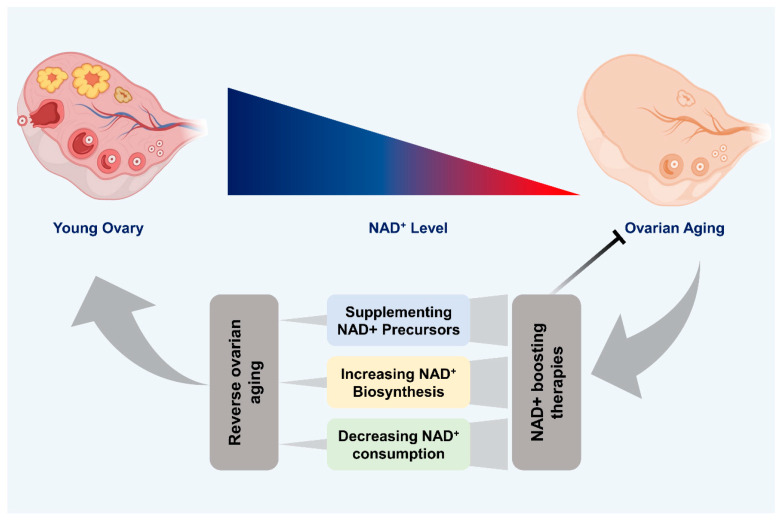
Therapeutic potential of NAD^+^ boosting. The figure depicts the age-related decline in NAD^+^ levels within the ovary. The young ovaries have sufficient NAD^+^ concentration, while its level decreases with age. This decline in NAD^+^ parallels the deterioration of ovarian function and fertility associated with aging. To address this NAD^+^ decline, the figure outlines three potential therapeutic interventions aimed at rejuvenating aging ovaries: NAD^+^ boosting through supplementation of its precursors, increasing NAD^+^ biosynthesis, and decreasing NAD^+^ consumption. These strategies hold promise for restoring NAD^+^ homeostasis, thereby restoring ovarian health and potentially extending reproductive lifespan.

## Data Availability

Not applicable.

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
