# Peer review of "A Molecular Perspective and Role of NAD+ in Ovarian Aging"

_ijms, 2024, doi:10.3390/ijms25094680_

Round 1
Reviewer 1 Report
Comments and Suggestions for Authors
In the paper “A molecular perspective and role of NAD+ in ovarian aging”, Mehboob Ahmed and co-authors reviewed and summarized the state-of-the-art on the role on NAD+ metabolism in ovarian aging. The topic presented in the manuscript is extremely interesting and relevant since the ovarian aging field is currently a rapidly expanding field with important implications for woman’s healthspan and lifespan. The manuscript is generally clearly written and includes most of the relevant publications to discuss the topic. However, I suggest the following points to be addressed prior to publication:
· The title of section 2 is: “molecular mechanisms underlying ovarian aging”, however, the section covers a wide range of topics that are not related to molecular mechanisms (e.g. aging) and that are more appropriate for the introduction section. In fact, molecular mechanisms are barely mentioned in this section.
· The authors mentioned PCOS throughout the manuscript, however, it is not clear its relevance for the discussion of ovarian aging. The authors should clarify this point.
· Some sentences are unclear and should be improved, for examples, lines 247-248. The authors should review the manuscript carefully.
· Please add reference to the figure for the section 4.2.1 (De novo pathway)
· Sentence in lines 270-273 is incorrect. While the De Novo and the Salvage pathways have the enzymes NMNATs in common, the last two steps of the pathways are not the same. Please correct this statement. Also, make sure you use same acronyms for the metabolites in the text and figures (e.g. NAaD vs NAAD).
· Lines 567-568. The reference mentioned demonstrated that NR is resistant to CD38 degradation. Please correct the statement or include another reference.
· Line 547. CD38 does not influence only nuclear Ca2+ homeostasis. Please correct this statement.
· Line 580. CD38 hydrolase activity also converts NAD+ into NAM and ADPR. Please review the paragraph about CD38 enzymatic activity for accuracy.
· Please carefully check the references as some of them are repeated (e.g. refs 63 and 153)
· Lines 599-602. It is unclear what “CD38 has been widely expressed in mice” means. Please clarify.
· Authors should mention ongoing clinical studies with NAD+ supplements in the context of ovarian aging/fertility.
· Lines 703-704. The authors should mention this recent study demonstrating the in vivo efficacy of 78c in improving mouse fertility: https://www.nature.com/articles/s43587-023-00532-9
· Any data on boosting NAD+ biosynthesis via NAMPT in the context of ovarian function to mention in section 5?
Comments on the Quality of English LanguageThe manuscript is generally clearly written, however, moderate editing is required to improve clarify in some sections (see comments to the authors).
Reviewer 2 Report
Comments and Suggestions for Authors
Dear Authors!
I read your manuscript very carefully and found it very interesting, well-written and -organized with many explanatory figures. I think you make clear the significance of NAD+ in the ovarian aging, considering its association with sirtuins (SIRT1-6), PARPs and CD38. I suggest your manuscript suitable for publication, but I would like to incorporate an important section in this review regarding the increase of NAD+ biosynthesis through nutrition. You can refer the food source of Nicotine acid (Vitamin B3), which can be converted to NAD+.
As a minor point: Write the titles in the Table 1 (see line 637)
Comments on the Quality of English LanguageEnglish language was fine.
